# Randomised controlled feasibility trial of the Active Communication Education programme plus hearing aid provision versus hearing aid provision alone (ACE to HEAR): a study protocol

Nicholas J Thyer,[1] Jude Watson,[2] Cath Jackson,[3] Louise Hickson,[4] Christina Maynard,[1] Anne Forster,[5] Laura Clark,[2] Kerry Bell,[2] Caroline Fairhurst,[2] Kim Cocks,[2] Rob Gardner,[6] Kate Iley,[7] Lorraine Gailey[8]

For numbered affiliations see end of article.

**Correspondence to**
Dr Nicholas J Thyer;
n.j.thyer@leeds.ac.uk

## ABSTRACT

**Introduction** Up to 30% of hearing aids fitted to new adult clients are reported to be of low benefit and used intermittently or not at all. Evidence suggests that additional interventions paired with service-delivery redesign may help improve hearing aid use and benefit. The range of interventions available is limited. In particular, the efficacy of interventions like the Active Communication Education (ACE) programme that focus on improving communication success with hearing-impaired people and significant others, has not previously been assessed. We propose that improved communication outcomes associated with the ACE intervention, lead to an increased perception of hearing aid value and more realistic expectations associated with hearing aid use and ownership, which are reported to be key barriers and facilitators for successful hearing aid use. This study will assess the feasibility of delivering ACE and undertaking a definitive randomised controlled trial to evaluate whether ACE would be a cost-effective and acceptable way of increasing quality of life through improving communication and hearing aid use in a public health service such as the National Health Service.

**Methods and analysis** This will be a randomised controlled, open feasibility trial with embedded economic and process evaluations delivered in audiology departments in two UK cities. We aim to recruit 84 patients (and up to 84 significant others) aged 18 years and over, who report moderate or less than moderate benefit from their new hearing aid. The feasibility of a large-scale study and the acceptability of the ACE intervention will be measured by recruitment rates, treatment retention, follow-up rates and qualitative interviews.

**Ethics and dissemination** Ethical approval granted by South East Coast-Surrey Research Ethics Committee (16/LO/2012). Dissemination of results will be via peer-reviewed research publications both online and in print, conference presentations, posters, patient forums and Trust bulletins.

**Trial registration number** ISRCTN28090877.

### Strengths and limitations of this study

► This will be the first study to evaluate the processes involved in delivering the Active Communication Education (ACE) intervention in a general practitioner referral pathway for new National Health Service hearing aid users.
► If the randomised controlled trial (RCT) is shown to be feasible then the data from this study will provide critical information that will inform the design of a larger RCT to determine the social, clinical and economic outcomes of the ACE in this important clinical pathway.
► The study is powered to allow the SD of the proposed outcome measures to be estimated with reasonable certainty to inform future sample size calculations.
► This study is limited to assessing the feasibility of RCT and ACE delivery processes. Ultimately, the test of whether the ACE intervention leads to long-term communication success, better hearing aid use, hearing aid benefit, quality-of-life and economic gains, will be tested in a future full-scale RCT.

## INTRODUCTION

Age-related hearing impairment is a major worldwide public health issue for ageing populations.[1] It is reported as the third most common chronic condition affecting approximately 328 million (91%) middle-aged and older adults,[2] over 10 million adults in the UK alone.[3 4] By the age of 70, 70% will have a mild or worse hearing impairment, progressively worsening with age.[5] Hearing impairment is commonly associated with reduced quality of life and well-being[6–8] including depression[9] and anxiety,[10] social isolation,[8] poor social interactions,[11–13] cognitive dysfunction,[14] increased risk of developing dementia and reduced emotional, behavioural and general social

well-being.[15] In addition, disability in these domains is often experienced by normally hearing significant others (SOs) living with hearing-impaired people.[16–18] Hearing impairment therefore represents an enormous burden on society and the economy.

In high-income countries, the most common treatment is to fit a hearing aid.[6] Despite strong evidence that hearing aid use is associated with reductions in hearing disability noted above,[6 19–21] hearing aid use is remarkably low.[22 23] It is estimated that up to 30% of UK adult hearing aid owners do not use them regularly or at all.[4 24–26] International studies support these data.[8 27–29] Cost implications for the National Health Service (NHS) are significant as they provide 80% of UK hearing aids,[25] fitting >300 000 new devices each year of which an estimated 20 000–120 000 are unused. Even with global advances in technology, fitting protocols and outcome measurements,[30] there is little evidence that NHS hearing aid use and the expected gains in benefit and quality of life have improved over the last decade[19 25] and treatment continues to focus primarily on technology.

A recent systematic review of additional treatment found very low quality evidence that self-management and service delivery interventions may be of benefit in auditory rehabilitation.[31] However, the authors found no studies that examined the effect of these sorts of interventions on hearing aid use.

Reasons for hearing aid non-use are complex.[32] Research has identified psychosocial factors important for successful aural rehabilitation including personal and societal attitudes to hearing-impairment[33] patient involvement in decision making[34–37] and expectations of benefit and communication success in a range of communication situations.[38 39] Key barriers and facilitators to successful hearing aid use have been identified as being related to expectations of benefit and meaningful participation in everyday life.[40 41] The WHO's International Classification of Functioning, Disability and Health[42] provides a functional description of difficulties (related to these expectations) experienced by hearing-impaired people and their hearing communication partners, for example, avoidance of difficult listening situations that lead to 'activity limitations' and 'participation restrictions'.

One intervention that is designed to reduce these limitations and restrictions is the Active Communication Education (ACE) programme (the focus of this paper). The ACE trains participants to develop solutions to specific difficult communication scenarios that commonly lead to their avoidance of or reduced participation in important activities. The effectiveness of ACE as an *alternative* intervention to a hearing aid has been evaluated and two small studies demonstrate ACE benefits in improving communication function and hearing-related quality of life.[43 44] ACE effectiveness as an *adjunct* to hearing aid fitting has not been evaluated, although there is some weak evidence that supports its use in this context,[43] that is, long-term improvements in using communication strategies being associated with hearing aid users.

A systematic review of group communication programme effectiveness conducted in 2005,[44] revealed just nine small and methodologically poor studies. The review reported weak evidence for short-term benefits related to reduced hearing disability; improvements in quality of life; hearing aid use and communication function[45] when interventions were delivered in concert with a hearing aid. The authors concluded that there was a clear need for large sufficiently powered randomised controlled studies to determine short-term and long-term benefits of adult communication rehabilitation group interventions as an adjunct to hearing aid fitting. Such a study has yet to be completed. Evidence that post-dates this 2005 review does little to change the situation providing only weak additional evidence for moderate gains in hearing loss-related quality of life for communication-based group interventions.[46] There are indications that group rehabilitation programmes like ACE have the potential to realise economic gains for service providers. For example, Abrams[47] estimated that a hearing aid plus a 4-week group rehabilitation programme reduced the treatment cost per quality-adjusted life-year (QALY) gained by more than half; the cost of implementing the rehabilitation programme was <6% of the total rehabilitation cost per patient. Even so, with no strong evidence that such interventions delivered as an adjunct to hearing aid fitting are clinically worthwhile, they are not routinely offered in public or private hearing healthcare sectors in the UK.

In summary, there is some low-quality evidence that ACE and similar programmes improve communication function and quality of life and these outcomes may be enhanced when delivered in conjunction with a hearing aid. Communication programmes involve a substantial commitment on the part of participants and those who run and pay for them. Recent evidence shows that it is effective and feasible to deliver ACE as an alternative intervention in Australia and Sweden.[43 48–50] There is now a need to establish whether reported clinical and economic benefits of ACE and communication programmes like ACE can be achieved in the context of NHS hearing aid provision. This protocol for The ACE to HEAR study (ACEto improve HEARing) is intended to deliver ACE to unsuccessful or struggling new NHS hearing aid users, 3 months postfitting, in order to assess whether a large RCT designed to evaluate the effectiveness of ACE in improving hearing aid benefit within the UK NHS is feasible.

## METHODS AND ANALYSIS

This protocol was developed and is reported according to the Standard Protocol Items for Randomised Trials statement.[51]

### Study aim

The aim of this study is to determine the feasibility of delivering a future, full-scale randomised controlled trial

(RCT) to evaluate ACE plus treatment-as-usual versus treatment-as-usual alone, within the UK NHS in two UK cities. Treatment-as-usual is defined as a referral from a patient's GP to audiology services to treat permanent hearing loss. It comprises up to two appointments for hearing aid fitting and a third face-to-face or telephone follow-up appointment.

## Study objectives
Objectives will evaluate ACE delivery and trial delivery processes.

### ACE delivery objectives
1. To assess ACE uptake rates, eligibility and acceptability of clinic location (between and within the two study sites) for participants and SOs.
2. To evaluate the level of ACE attendance and retention among participants randomised to the ACE arm of the study.
3. To assess acceptability of ACE with participants, SOs and audiologists.
4. To assess capability, capacity and willingness of audiology departments to support delivery of ACE within existing services.
5. To assess intervention fidelity of delivering ACE.

### Trial delivery objectives
1. To assess RCT recruitment rates, evaluate the randomisation process and time to accrue ACE groups.
2. To assess the acceptability of study processes to participants, SOs and audiologists (ie, those related to recruitment, the feasibility of identifying struggling hearing aid users, randomisation process, data collection, measurement of ACE fidelity and acceptability).
3. To explore patient-reported outcome measures (PROMs) and estimate likely SD, including quality-of-life tools (EQ-5D-5L; Short-Form 36 (SF-36)) and a bespoke healthcare resource use/acceptability/utility questionnaire for use in an intended full-scale RCT.

## Study design
This study commenced on 1 February 2017 and is of 24 months duration. The study design is a randomised controlled, open feasibility trial with embedded economic and process evaluations delivered in one audiology department in each of two UK cities. The design of the trial is shown in figure 1.

## Study setting
Study sites for this feasibility study are the Audiology Departments at York Hospital (YH), York Teaching Hospitals NHS foundation Trust and the Bradford Royal Infirmary (BRI), Bradford Teaching Hospitals NHS Foundation Trust.

## Study population
The study population will consist of adult patients aged 18 years or over, receiving treatment-as-usual delivered in one of the two participating centres. They will be considered potentially eligible if all of the following eligibility criteria apply at their 3-month postfitting follow-up appointment.

## Inclusion criteria
a. Moderate or less than moderate benefit, defined by International Outcomes Inventory for Hearing Aids (IOI-HA) question 2.[52 53]
b. Hearing impairment: pure-tone average better ear thresholds at 500, 1000, 2000 and 4000 Hz of >25 dB HTL.
c. No significant self-reported history of neurological impairment.
d. Willing to provide written informed consent.
e. Able to provide written informed consent.
f. Able to take part in the intervention by understanding and using spoken English.
g. Able to self-complete the English language outcome measure tools.
h. The following inclusion criteria for SOs will be assessed:
   a. A spouse or other family member who lives with or is a carer for a patient recruited to the study.

## Exclusion criteria
a. Severe or profound bilateral hearing impairment. Pure-tone better ear average thresholds measured at 500, 1000, 2000, and 4000 Hz of >85 dB Hearing Threshold Level (HTL), since experience[48] suggests this group of patients may struggle to effectively participate in the intervention setting.
b. Significant ongoing ear-related health or mental health issues that, in the audiologist's or associate audiologist's professional opinion, would preclude hearing aid fitting or attendance at ACE sessions.
c. Unable or unwilling to give written informed consent.
   In addition, SOs will be excluded if they are unable or unwilling to give written informed consent.

Patients who do not have a SO or family member able to attend the ACE sessions are still eligible to participate in the study.

## Sample size calculation and recruitment
As this is a feasibility study, the main purpose is to assess the acceptability and feasibility of conducting this study, with a view to designing and conducting a future full-scale trial.[54] Six ACE groups are planned with up to seven patients in each (five minimum). This leads to a maximum sample size of 44 patients and up to 44 SOs for the ACE intervention arm, and 44 in the control arm (n=88). This sample size will allow the SD of the proposed outcome measures to be estimated with reasonable certainty to inform future sample size calculations.[54] Collectively, the two study sites fit approximately 4300 hearing aids a year and based on their experience we estimate 10% will require extra help at follow-up and be eligible for inclusion. Recruitment of patients commenced on 1 April 2017 and is now underway. The recruitment window is currently planned to end on 28 February 2018 and there is potential to extend this phase until 30 April 2018 if necessary.

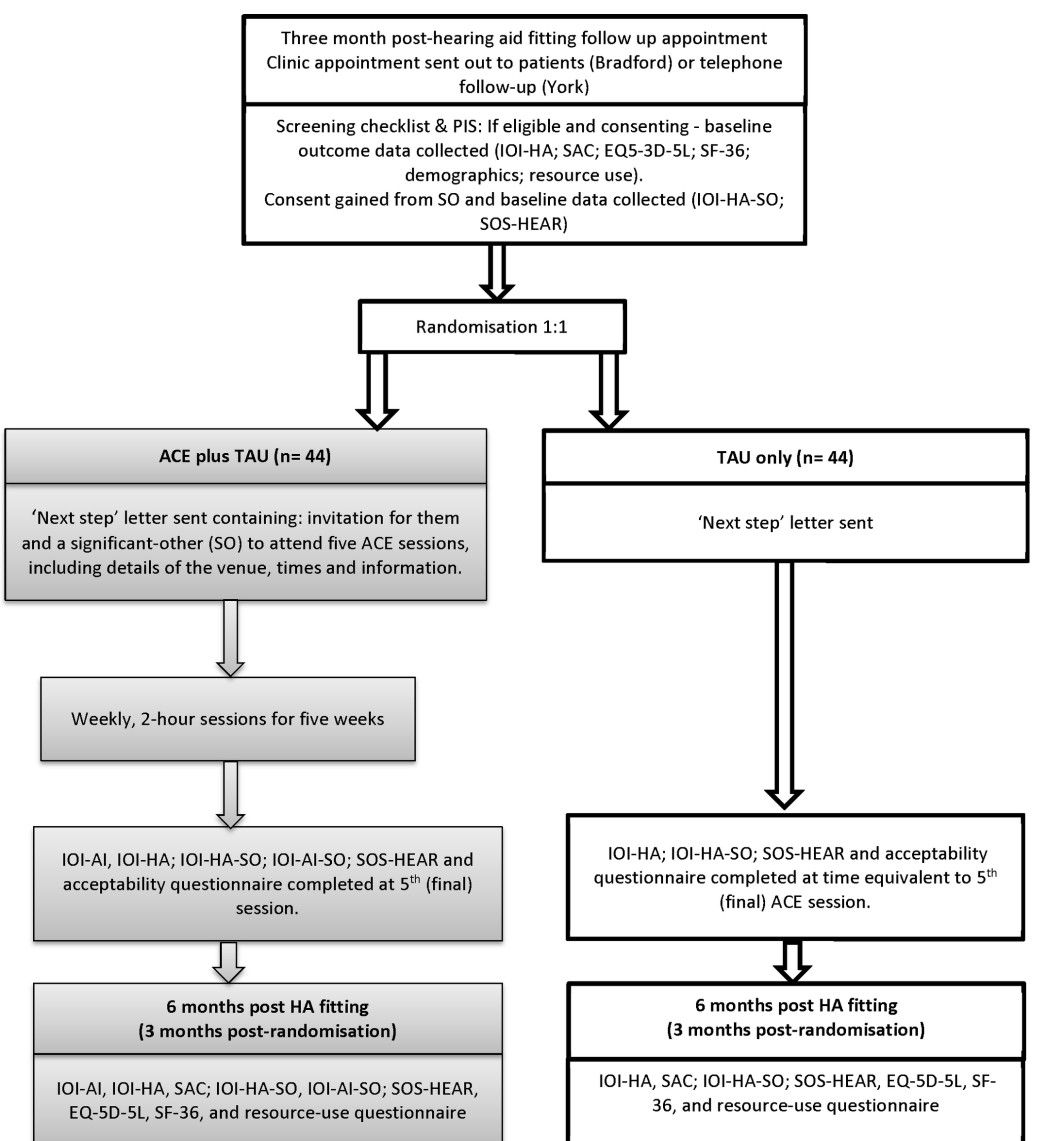

**Figure 1** ACE to HEAR study flow diagram. The diagram was developed using Consolidated Standards of Reporting Trials guidelines (http://www.consort-statement.org/) and indicates the main processes of the trial, their relationship with the outcome measures and their timing. ACE, Active Communication Education; IOI-HA, International Outcomes Inventory for Hearing Aids; IOI-HA-SO, IOI for hearing aids: version for SO; IOI-AI, International Outcomes Inventory for Alternative Interventions; IOI-AI-SO, IOI for Alternative Interventions: vwesion for SO; SAC, Self-Assessment of Communication; SF-36, Short-Form 36; SO, significant other; SOS-HEAR, Significant Other Scale for Hearing Disability.

## Randomisation

Eligible, consenting patients from the same study site who have completed baseline assessments will be randomised by a remote, centralised randomisation service (provided by York Trials Unit) in batches of 10–14 (intervention:control ratio of 1:1) using block randomisation in a single large block per batch. Following randomisation, a letter outlining the next steps will be sent to participants. For those allocated to the ACE arm, this will contain an invitation for them and their SO to attend five ACE sessions, including details of the venue, times and information.

## Blinding

By the nature of the interventions used within this study, blinding of the participants and facilitator is not possible.

The research team responsible for data analysis and reporting will be blinded where possible.

## Intervention allocation

Consenting participants will be randomly allocated to either:

► *treatment-as-usual:* participants randomised to receive treatment-as-usual will continue to receive usual care delivered by their service.
► *ACE plus treatment-as-usual:* participants randomised to receive the ACE plus treatment-as-usual, will attend five 2 hours weekly sessions of the ACE programme, developed in Australia[48] and will continue to receive usual care delivered by their service.

The ACE programme will be delivered as described in the published ACE manual[55] by a trained facilitator to groups of hard of hearing people and SOs where possible. The same audiologist facilitator will deliver ACE to all groups.

ACE consists of six modules based on everyday communication activities known to be problematic for hard of hearing adults: module 1: communication needs analysis; module 2: conversation in background noise; module 3: conversation around the house; module 4: communication with difficult speakers; module 5: listening to other signals and module 6: listening to public-address systems.

Through the use of demonstrations, practical exercises, discussion and problem solving, participants and SOs will learn individual problem-solving skills which can be applied in a range of novel situations and discuss the use of communication strategies, lip reading skills, clarification skills and assistive technology.

### Intervention fidelity

Assessment of the fidelity to the ACE training of the audiologist facilitator and the delivery of the ACE programme will follow guidance from the NIH Behaviour Change Consortium,[56] and conducted in a pragmatic approach mirroring the way fidelity of an educational programme, such as ACE, would be realistically assured in an NHS context. The trainers will reflect on the training sessions; fidelity of ACE delivery will be *facilitated* by supervision of the audiologist for the first session, with feedback and self-reflection used to address any issues; fidelity of ACE delivery will then be *assessed* via a self-monitoring form recording on a 1–4 scale (poor–good) of their adherence to the ACE manual for each module across all sessions.

### Intervention compliance

Measuring compliance is challenging in complex interventions such as this, where there are a number of interacting elements (such as the influences of the ACE facilitator, participants' significant other, a varied selection of ACE module topics worked on and each individuals' perceptions of their (or their SO's) disability and treatment benefit). The intervention to be delivered is defined in the ACE manual[55] and our understanding (measurement) of what is actually delivered will be informed by the fidelity measures above, weekly attendance logs and weekly session records. ACE participants and SOs will self-report goal setting for each module and the number of goals achieved during the programme, the number of completed exercises (homework) will be reported providing an additional measure of the level of engagement or compliance with the programme that is focused on the participant and their SOs.

### Concurrent treatments

Any additional treatments identified will be available to both arms when necessary (eg, hearing aid repairs, replacement batteries, etc). The bespoke resource use questionnaire (see below) will allow us to monitor additional treatment accessed during the study.

### Outcome measures

Figure 1 shows the main processes of the trial, their relationship with the outcome measures and their timing. Data will be collected at baseline (prior to randomisation); during all ACE sessions and after ACE session 5 for ACE participants and at an equivalent time for treatment-as-usual participants; and approximately 3 months post randomisation (6 months post hearing aid fitting).

The feasibility and the potential for a future large-scale study design (the study's aim) will be measured by recording and evaluating:

#### ACE delivery outcomes

1. *Attendance*: Attendance rates of participants and SOs at ACE sessions will be measured and who attends ACE session with the participant will be recorded (objective 2).
2. Fit of ACE with existing variations in service delivery models: comparison of attendance at ACE delivered at different study sites and their satellite clinics and the effect of using telephone or face-to-face follow-up appointments will be recorded (objectives 1 and 4).
3. Can ACE be delivered as intended in the ACE protocol? This will be assessed through: (i) time taken to train the ACE facilitator successfully; (ii) number of ACE goals achieved by participants and (iii) facilitator's adherence to the ACE protocol (fidelity) (objective 5).
4. *Acceptability*: bespoke acceptability questionnaire regarding study processes, designed to explore participant and SO thoughts regarding the study including; ACE session organisation, session content, being approached and informed about the study and completing the questionnaires. The questionnaire is administered to participants and SOs at the final ACE session and at a comparable time for the questions for treatment-as-usual arm (objectives 3 and 4).

#### RCT delivery outcomes

1. *Recruitment*: number of follow-up cases in GP referred pathway; number of follow-up cases in GP referred pathway struggling with their hearing aid; number of and reasons for exclusions; number of patients who decline to participate and reason for declining; number who miss ACE intervention window (ie, unable to attend an ACE group within 1–3 weeks after randomisation); number given an appointment for an ACE group session and number of consented participants who fail to attend ACE sessions (objectives 6 and 7).
2. *Allocation*: time taken to recruit and logistics of recruiting an optimally sized and located ACE group; time ACE started after randomisation (ACE intervention window); (objective 6).
3. *PROM data*: completion of the outcome measures below at each time-point will be recorded as well

International Outcomes Inventory fo Hearing Aids as extent of missing data within each outcome measure (objective 8).

## Patient-reported outcome measures

▶ *International Outcomes Inventory fo Hearing Aids (IOI-HA)*[52]: a seven-item questionnaire designed to evaluate the effectiveness of hearing aid treatments. The domains covered are: daily use; benefit; residual activity limitations; satisfaction; residual participation restrictions; impact on others and quality of life. Responses are assigned a value from 1 to 5 and values summed. Higher scores indicate a more favourable outcome.

▶ *International Outcomes Inventory for Alternative Interventions (IOI-AI)*[57]: a version of the IOI designed for use for non-hearing aid-based interventions such as ACE, covering the same domains as the IOI-HA.

▶ *Self-Assessment of Communication*[58]: designed to measure the effect of hearing loss and hearing aid outcomes. This 10-item instrument covers questions about communication problems using a Likert scale ranging from 1 ('almost never') to 5 ('practically always'). A percentage score is calculated by multiplying the raw score by 2, subtracting 20 and multiplying by 1.25.

▶ EQ-5D-5L[59 60]: a standardised generic instrument for describing and valuing health in terms of five dimensions (mobility, self-care, usual activities, pain/discomfort and anxiety/depression) using five levels of severity. Overall health on the day is also rated by the respondent on a 0–100 vertical visual analogue scale.

▶ *SF-36*[61]: a generic health measure with 36 items assessing eight health concepts: physical functioning; role limitations due to physical problems; general health perceptions; vitality; social functioning; role limitations due to emotional problems; general mental health and health transition.

The following will be completed by participants' SOs only:

▶ *IOI-AI: version for significant others*[56]: a version of the IOI designed for use with SOs and non-hearing aid-based interventions covering the same seven domains as the IOI-HA.

▶ *IOI for hearing aids: version for SOs*[56]: an extension of the IOI-HA for use with the SO covering the same seven domains as the IOI-HA.

▶ *Significant other scale for hearing disability*[16]: a 27-item self-report tool, which assesses third-party disability in spouses of older people with hearing impairment. It measures the effects of hearing impairment on the SO in the following domains: changes to communication; communication burden; relationship changes; going out and socialising; emotional reactions to adaptations; concern for partner. It uses a 5-point response scale: 0=no problem to 4=a complete problem.

The feasibility of collecting postal questionnaire data at each time point will be evaluated. Table 1 shows the data collection schedule.

## Screening and enrolment

Patients attending Audiology Clinics at YH and BRI will be approached to take part. Treatment-as-usual provided at the posthearing aid fitting follow-up will be according to site and therefore the recruitment process will vary slightly at each site:

▶ *York*: patients will be followed up via a telephone interview. Eligibility will be checked during telephone interview and from medical records. Details of eligible and interested patients will, with their permission, be passed onto a non-ACE researcher who will post out a patient information sheet (PIS) and conduct a telephone follow-up call a few days later to see if still interested. Contact information is provided in the PIS so that the patient has opportunity to ask questions regarding the study. If willing to participate, informed consent and baseline questionnaire will be completed by post. SOs of patients who are recruited at York will receive a SO-specific PIS, consent form and baseline questionnaire by post to be returned in a freepost envelope.

▶ *Bradford*: patients will be offered a face-to-face follow-up appointment 3 months post hearing aid fitting. Eligibility will be checked at this appointment and from medical records. Details of eligible and interested patients will be passed onto a non-ACE researcher who will provide a PIS and discuss the study. The patient will have the opportunity to ask questions and if willing, provide informed consent and complete a baseline questionnaire. If further consideration is required, the patient will be contacted by telephone call a few days later to see if still interested. In addition, if an SO attends the appointment with a patient, they will be provided with a SO-specific PIS, consent form and baseline questionnaire. Otherwise the documentation will be given to the patient to pass on to their SO or posted out.

We will monitor the proportion of patients referred to the treatment-as-usual pathway during the study recruitment who subsequently do not attend or are not contactable by telephone for their post-HA fitting follow-up in order to estimate how many referrals may potentially be lost to recruitment. We will liaise with audiologists to identify reasons for non-attendance where possible.

## Data collection and management

All data for the participant outcome measures will be collected by self-completed questionnaires returned by post or in secure boxes within the audiology clinics. Participants and SOs who fail to return their postal questionnaires will receive one reminder letter. Participants may also be asked to complete a questionnaire over the telephone, or asked to provide any missing data if required. Data from these paper forms will then be entered into a

**Table 1** Data collection schedule Data are collected approximately 3 months posthearing aid fitting (baseline); at each ACE session 1–5 for the intervention arm and at a time equivalent to ACE week 5 for the treatment-as-usual arm; and approximately 6 months posthearing aid fitting

| Study period | Recruitment | Allocation | Postallocation | | | | | |
| --- | --- | --- | --- | --- | --- | --- | --- | --- |
| | | | ACE week 1 | ACE week 2 | ACE week 3 | ACE week 4 | ACE week 5 | |
| Time point | Baseline | 0 | | | | | | 6 months |
| **Recruitment** | | | | | | | | |
| Eligibility | • | | | | | | | |
| Informed consent | • | | | | | | | |
| Optional qualitative study consent | • | | | | | | | |
| Allocation | | • | | | | | | |
| **Assessments** | | | | | | | | |
| Demographics | • | | | | | | | |
| IOI-HA | • | | | | | | • | • |
| SAC | • | | | | | | | • |
| EQ-5D-5L | • | | | | | | | • |
| SF-36 | • | | | | | | | • |
| Resource use | • | | | | | | | • |
| IOI-AI* | | | | | | | • | • |
| ACE participant attendance* | | | • | • | • | • | • | |
| ACE SO attendance~ | | | • | • | • | • | • | |
| IOI-AI-SO*† | | | | | | | • | • |
| IOI-HA-SO† | • | | | | | | • | • |
| SOS-HEAR† | • | | | | | | • | • |
| Acceptability questionnaire | | | | | | | • | |
| Qualitative interviews (participant and SO) | | | | | | | • | |
| Qualitative interviews (audiologists) | | | | | | | | • |

*ACE arm only.
† Significant others only.
ACE, Active Communication Education; IOI-AI, International Outcomes Inventory for Alternative Interventions; IOI-HA, International Outcomes Inventory for Hearing Aids; IOI-HA-SO, IOI for hearing aids: version for SO; SAC, Self-Assessment of Communication; SF-36, Short-Form 36; SO, significant others; SOS-HEAR, Significant Other Scale for Hearing Disability.

master database for the trial using either optical scanning techniques or entered manually.

Participants may withdraw from all or any aspects of the study without influencing their future care or treatment. A brief update of how the study is progressing will be sent out in order to maintain participant engagement with the study.

All information collected during the course of the trial will be kept strictly confidential. Information will be held securely on paper and electronically at York Trials Unit. All trial data will be identified using a unique trial identification number. Analytical datasets will not contain any identifiable information. Data will be archived for a period of at least 10 years following the end of the study.

### Statistical analysis

A single analysis will be conducted at the end of the trial using Stata V.13 or later. Data summaries and analyses will inform the design of a full-scale RCT of the intervention. Baseline data will be summarised by trial arm, using descriptive statistics for continuous data (mean, SD, median, minimum, maximum, number missing) and counts and percentages for categorical data. Recruitment rates will be reported monthly and overall, and by site. The flow of participants through the trial will be detailed in a Consolidated Standards of Reporting Trials flow diagram and referral, consent and attendance rates will be summarised overall and by site using counts, percentages and 95% CIs.

The number of ACE sessions attended will be summarised alongside any SOs who attended the sessions. Summaries will be provided overall, by site/clinic and by whether follow-up appointments were made as single or block booking. Acceptability data using Likert scales

at 6 months, for participants, SOs and audiologists will be summarised separately using summary statistics and presented graphically using bar charts, by trial arm.

The number of participants withdrawing from the ACE intervention and/or the trial and any reasons for withdrawal will be summarised.

The time taken to train audiologists to deliver ACE and the number of ACE goals achieved by participants will be summarised. Fidelity scores (from self-monitoring forms) will be summarised overall and by session. The proportion of training and ACE intervention delivered as intended, as well as any adaptations to training/intervention will be reported. Variations in dose of ACE intervention will be measured through ACE attendance and attrition data.

Questionnaire return rates at each time point will be presented overall and by trial arm. PROMs at each time point will be summarised descriptively overall, by trial arm, and by ACE group for participants allocated to receive the ACE intervention. SD will be presented with 80% CIs to inform future sample size calculations. The proportion of participants at the floor and ceiling of each measure, at each time point, will be reported along with the standardised response mean (SRM) to measure the sensitivity of each questionnaire to detect change. The SRM will be calculated as mean change in scores or values divided by the SD in change scores.[62] Questionnaire completion times (from self-report) will be summarised as a consideration for instruments going into the full-scale evaluation. Missing data will be reported as a proportion of the total expected data set for each measure and will inform feasibility.

### Qualitative data
Following delivery of all the ACE intervention sessions:
► The facilitator will be interviewed to explore the training and implementation process and their experiences of delivering ACE including barriers/facilitators to adhering to the ACE protocol.
► Up to three audiologists from both study sites (up to six in total) will also be interviewed, exploring the capability, capacity and willingness of their audiology departments to support the ACE study within their existing services. The acceptability of study processes will also be explored.
► A sample of 10–12 participants in the ACE intervention arm and four participants in the control arm (with their SOs if willing) will also be interviewed as soon as possible after the completion of the ACE sessions. We will purposively select participants to ensure a mix of those with good/poor hearing aid outcomes (measured at the fifth and final ACE session and equivalent control arm time point, see figure 1 for outcome measures) as well as ensuring we include some participants who dropped out of the sessions/study, ensuring a wide range of views are collected. Control arm participants will include those who have dropped out from the study where possible, allowing us to explore reasons for this. SOs will be interviewed

as part of a dyad with the participant. Semi-structured interviews will explore the acceptability of the ACE (eg, venue, timing, content), its perceived impact (reflecting on hearing aid outcomes) with ACE intervention arm participants; and views on study processes (eg, recruitment, outcome measures and timing) with ACE intervention and control arm participants.

Interviews will be audio-recorded, transcribed verbatim and analysed (with NVivo-11) by the research team led by CJ, using the Framework approach,[63] which is particularly useful for analysing qualitative data in a pragmatic yet systematic way, where theoretical development is not needed. The steps are familiarisation, construction of a thematic framework, indexing and charting the data, mapping and interpretation.

### Economic analysis and quality of life data
A full cost-effectiveness analysis will not be conducted as this is a feasibility trial, thus the study is not powered to detect significant differences.

The costs of implementing the ACE intervention will be estimated and the potential resource implications versus usual care will be explored. Resource use will be summarised by resource use type (eg, GP appointments, outpatient appointments) and appropriate unit costs to be applied to each resource use type will be identified. These will be sourced from a combination of local costings and national databases.[64]

The costing approach will take a broad analytical perspective accounting for NHS costs and for those observed by patients, although this cost will be presented separately. It is anticipated that additional resources used in the ACE intervention arm will largely be NHS staff time and travel/time for patients and SOs, patients and SOs use of primary and secondary NHS care, any private treatments attended, whether related to their hearing or for any other reason, changes to medication and employment or recreational activities. A draft resource use questionnaire based on these anticipated additional resources was designed for this study. The questionnaire will be developed further and tested during the feasibility trial to ensure that all relevant and necessary data can be collected to establish a reliable and valid tool with which to capture resource use for a future full economic evaluation.

Methods to estimate an incremental cost-effectiveness ratio for the ACE intervention versus treatment-as-usual alone in terms of QALYs will be explored. No health-related quality of life assessment tool is currently sufficiently sensitive to all populations, and in the field of hearing health, there has been limited research to identify the most effective tool. In the UK, NICE advocates the used of the EQ-5D for generating QALYs, although it is acknowledged that this is not always the most sensitive tool for particular populations for whom the majority of its dimensions may not apply.[65] In the US analysis of a 4-week rehabilitation programme noted above,[47] the SF-36 was used to generate QALYs rather than the EQ-5D, with the SF-36 showing a reduction in the cost per QALY in favour of the intervention. For the present

feasibility trial, both assessments will be used to elicit QALYs and a comparison will be made between the outcomes of the two measures. This will enable a decision to be made as to which tool would be most appropriate in the full-scale trial.

The feasibility work will also be used to help to identify any patterns of missing data and any issues relevant for sensitivity analysis, which will influence statistical plans for dealing with imprecision and other uncertainties in the full RCT. For example, data can be bootstrapped to account for the expected skewness evident in economic cost data. The data collected as part of this feasibility study will be used to inform subsequent pretrial modelling.

### Adverse events

Risks within this study are considered to be minimal. It is considered highly unlikely that the ACE intervention arm will suffer any adverse consequences as a result of receiving the ACE plus treatment-as-usual. Nevertheless, interviews with ACE participants, the ACE facilitator and ACE questionnaire data will be used to monitor this eventuality.

### Trial monitoring and oversight

Due to the low risk nature of this trial, approval has been obtained to set up one independent steering and monitoring committee to undertake the roles traditionally undertaken by the Trial Steering Committee and Data Monitoring and Ethics Committee. Regular meetings of a Study Management Group will take place to oversee the progress of the study and review recruitment. We will establish a Project Advisory Panel (PAP) with between two and four hard of hearing adults or hearing spouses that will meet a minimum of five times over the duration of the project. The PAP is a group of patients, service users, carers and lay members of the public whose role is to support and advise the Study Management Group on all aspects of the study's progression and management.

### Patient and public involvement

Three patient and public involvement (PPI) activities informed the development of this application. First, a funded public engagement event about public perceptions of hearing impairment was held at the Thackeray Medical Museum in Leeds. This event helped to inform the research question. Participants identified a need for wider availability of treatments additional to hearing aids and that non-technological interventions for hearing-impaired people were a priority. Delegates identified communication education as a useful addition to hearing aid use for many hearing-impaired people and their family members. This feedback informed the study design in the following way: the choice of an interactive communication-based intervention rather than an informational one; the need to ensure that routine practical information about hearing aids and hearing impairment is delivered consistently and checked after fitting.

Second, a focus group was held to consult with service-users on the proposed research question, study design and intervention delivery. Four participants were asked to discuss (a) study information and consent procedures, (b) factors that might encourage or discourage their participation in the proposed study such as the burden of the intervention, (c) types of communication scenarios that are important to them, (d) factors that might motivate them to be more active communicators. The outcomes informed our recruitment strategy to maximise interest, commitment and recruitment rates. The group's views helped develop study information sheets and operational components of delivering ACE.

Third, the charity Hearing Link, who have extensive experience of PPI and managing and delivering group interventions of this type, were consulted about involving public and patients in operationalising and delivering ACE. Patients and service users and carers are involved in the conduct of this study as active members of the PAP.

We will present the findings of this study in patient forums, Trust bulletins and PPI activities including newsletters and public interest groups who work and support older adults with hearing impairment.

### Data monitoring and management

Information relating to study participants will be kept confidential and managed in accordance with the Data Protection Act, NHS Caldicott Guardian, Research Governance Framework for Health and Social Care and the Research Ethics Committee approval.

Participant details will be stored on a secure password-protected server located at the University of York, for the purposes of assisting in follow-ups during the study. All paper data collected from participants will be maintained in a safe secure environment at York Trials Unit. Paper records will be identified using identifiers rather than personally identifiable information. Analytical datasets will not contain any identifiable information.

The confidentiality of the participants, SOs and audiologists interviewed during their qualitative interviews will be ensured by assigning a unique identification code to electronic sound files and transcripts of interviews, known only to the qualitative researcher and appropriate members of the research team. Any quotes published will be anonymous further protecting participant confidentiality.

### ETHICS AND DISSEMINATION

Since the study started in February 2017, three Health Research Authority (HRA)-approved amendments were added to the protocol and are included in the final version reported here:

1. Revised the fidelity check tool. This was considered a non-substantial amendment.
2. Remove inclusion criteria of <3 hours hearing aid use a day and adjusted inclusion criteria to include moderate benefit. We also gained approval to distribute study information flyers to study site staff and patients. These were considered substantial amendments.

3. Developed a study information flyer to be sent to patients not contactable by telephone for follow-up at the York study site. This was considered a non-substantial amendment.

The proposed study will be conducted in accordance with the MRC Guidelines on Good Clinical Practice in Clinical Trials.

The results from this study will be submitted to the funders, peer-reviewed journals, presented at relevant meetings/conferences and within the participating and other audiology departments. We also intend to present the findings of this study in patient forums, Trust bulletins and PPI activities including newsletters and public interest groups who work and support older adults with hearing impairment.

## CONCLUSION
This will be the first RCT of this type of group communication programme in the context of a public health service and as an adjunct to hearing aid fitting. The impact of this study will ultimately be realised by a larger fully powered RCT designed to determine the effectiveness of the ACE intervention in improving hearing aid benefit for hearing aid users within the NHS GP referral for a hearing aid pathway in the UK. The outcomes of this study will inform such a RCT.

The feasibility study will be deemed successful if:
1. Seventy per cent of recruitment targets attained for all research components.
2. Study consent/retention rates and proposed sample sizes, indicate delivery of the full RCT is plausible within a 5-year study period.
3. Ninety per cent of ACE groups of five to seven consented participants formed within the intervention window with participants attending three of five sessions.
4. Economic, acceptability, outcome measure and fidelity evaluation data successfully collecte
5. Participants, SOs and audiologists evaluate acceptability of the ACE and RCT positively.

(Measures with over 10% missing data maybe modified/replaced prior to the main trial).

The criteria for success will result in one of following outcomes:
1. stop: full-scale RCT not be feasible in NHS setting;
2. continue: feasible with modifications;
3. continue: feasible with no modifications, close monitoring;
4. continue feasible as is.

## Author affiliations
[1]Leeds Institute of Cardiovascular and Metabolic Medicine (LICAMM), University of Leeds, Leeds, UK
[2]York Trials Unit, Department of Health Sciences, University of York, York, UK
[3]Valid Research Ltd, Wetherby, UK
[4]School of Health and Rehabilitation Sciences, The University of Queensland, Brisbane, Queensland, Australia
[5]Leeds Institute of Health Sciences (LIHS), University of Leeds, Leeds, UK
[6]Audiology Department, Bradford Royal Infirmary, Bradford, UK
[7]Audiology Department, York Hospital, York, UK
[8]Hearing Link, Eastbourne, UK

**Acknowledgements** The authors would like to thank Catherine Hewitt[2] for helpful statistical advice and comments on an earlier draft of this paper and members of the Project Advisory Panel (PAP), Emmanuelle Blondiaux-Ding, Philip Le Mare and Shona Hudson for their invaluable advice on many aspects of the project.

**Contributors** NJT led on the conception, design and writing of the study and study protocol with substantial contributions to the design, writing, critical review of intellectual content and final manuscript approval from JW, CJ, KC, AF, LH, LC, KB, CF, CM,RG, KI and LG. All authors agree to be accountable for their work. As Principle Investigator, NJT takes overall responsibility for the work. KC provided statistical expertise in the study design and development stages of the project and the protocol. CF provided further essential statistical advice and expertise on the study protocol. JW, CJ and LC made substantial contributions to the trial design and management. AF was involved in all aspects of the study and LG provided particular input to PPI. KB was specifically responsible for the health economic aspects of the study design and KI and RG were responsible for aspects specific to Trust's service delivery, providing expert clinical support.

**Funding** Sponsored by Bradford Teaching Hospitals NHS Foundation Trust are the sponsors on behalf of the funder, who is the NIHR as stated below. This paper presents independent research funded by the National Institute for Health Research (NIHR) under its Research for PatientBenefit (RfPB) Programme (Grant Reference Number PB-PG-0215-36147).

**Disclaimer** The views expressed are those of the author(s) and not necessarily those of the NHS, the NIHR or the Department of Health and Social Care.

**Competing interests** None declared.

**Patient consent** Not required.

**Ethics approval** Ethical approval has been granted by South East Coast-Surrey Research Ethics Committee. IRAS project ID 204072 (rec reference 16/LO/2012) and HRA approval obtained.

**Provenance and peer review** Not commissioned; externally peer reviewed.

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
