## [Reviewer comments · BMJ Open]

ARTICLE DETAILS

TITLE (PROVISIONAL)	A RANDOMISED CONTROLLED FEASIBILITY TRIAL OF THE ACTIVE COMMUNICATION EDUCATION PROGRAMME PLUS HEARING AID PROVISION VERSUS HEARING AID PROVISION ALONE (ACE TO HEAR): A STUDY PROTOCOL.
AUTHORS	Thyer, Nicholas; Watson, Jude; Jackson, Cath; Hickson, Louise; Maynard, Christina; Forster, Anne; CLARK, LAURA; Bell, Kerry; Fairhurst, Caroline; Cocks, Kim; Gardner, Rob; Iley, Kate; Gailey, Lorraine

VERSION 1 – REVIEW

REVIEWER	Vinaya Manchaiah Lamar University, USA
REVIEW RETURNED	01-Feb-2018

GENERAL COMMENTS	The manuscript presents a study design of a feasibility study. The manuscript is well structured and well presented. The study is much needed in this area. However, following minor issues need to be addressed before this manuscript can be considered for publication.  ♣ Differentiate the primary and secondary outcome measure. ♣ The study focuses more on “process” rather than “outcome”. There is a manuscript in the current issue of IJA about process evaluation of RCT. This may be a useful reference for this study. ♣ The data analysis section should include information about “missing data handling”. ♣ The potential study limitations of the proposed study should be highlighted.
---

REVIEWER	Helen Henshaw NIHR Nottingham Biomedical Research Centre, University of Nottingham, UK
REVIEW RETURNED	15-Feb-2018

GENERAL COMMENTS	This manuscript reports the protocol for a feasibility study that has already been granted ethical approval and is approaching the anticipated end-date for recruitment (28.02.18). The study is generally well described. Thus, my comments are limited to the clarity of the protocol itself. My first query relates to the description of the feasibility study type: P2 L30: The authors state ‘This will be a pragmatic, randomised controlled...’ Pragmatic trials are designed to test interventions in the full spectrum of everyday clinical settings to maximise applicability and generalisability (e.g. Treweek & Zwarenstein, 2009). Thus,
--

	pragmatic in this context suggests the recruitment of any new adult client receiving hearing aids. Yet, the feasibility study (and ensuing trial) limit inclusion based on pure tone audiometric criteria (mild and moderate, not severe or profound). I therefore suggest the authors remove the word pragmatic. The authors should also provide justification for the inclusion criteria limit on PTA within the methods (I expect this is related to the anticipated benefit of the intervention for those two subgroups, based on prior literature?) My second point relates to qualitative analyses: P19 L37: 'Interviews will be audio recorderd...' It is not clear who will be responsible for conducting the interviews and analysing/interpreting the data. P20, L16: As recruitment is well-underway, are the authors able to specify with more certainty what will be included on their HEA cost-form? i.e. what HE data they are collecting? P21, L25: What is the role of the PAP? Explain. P22, L9: 'Amendments' – this section is not really needed and could be deleted (if the authors really do wish to keep it, it should appear after ethics approval is stated). Final comment – the authors should specify a clear timeline, duration and end-point for the feasibility study, and provide full details of the criteria that will be used to judge whether or not the future RCT is feasible.
--	--

VERSION 1 – AUTHOR RESPONSE

Response to reviewers comments on: Manuscript ID bmjopen-2018-021502 entitled "A randomised controlled feasibility trial of the Active Communication Education programme plus hearing-aid provision versus hearing aid provision alone (ACE To HEAR): A study protocol."

Reviewer 1.

We thank this reviewer for their helpful comments and have addressed them in the revised manuscript as indicated in our response below.

1. Differentiate the primary and secondary outcome measure.

We considered differentiating outcomes for this feasibility study in this manner but decided against it. Since the outcomes relate to ACE delivery and RCT processes, it is clearer to group the outcome data under these headings rather than the more traditional primary and secondary outcome headings. In fact this is the approach taken in the paper this reviewer has kindly highlighted in point 2 below. We have not therefore changed the manuscript in this respect.

2. The study focuses more on "process" rather than "outcome". There is a manuscript in the current issue of IJA about process evaluation of RCT. This may be useful reference for this study.

We thank this reviewer for bringing this interesting article to our attention. Although we have not used it to revise our protocol manuscript, we feel sure it will help inform the remaining work on this study and any future trial in this area.

3. The data analysis section should include information about "missing data handling".

We have added the following text to P18L21-22 of the revised manuscript:

Missing data will be reported as a proportion of the total expected data set for each measure and will inform feasibility.

Since the analysis in this study is descriptive, statistical handling of missing data is not elaborated further.

4. The potential study limitations of the proposed study should be highlighted.

We have clarified the limitation of this feasibility study on P3L9 of the revised manuscript adding the following text:

“This study is limited to assessing the feasibility of RCT and ACE delivery processes”.

Reviewer: 2

We also thank this reviewer for their helpful comments and have addressed them in the revised manuscript as indicated in our response below.

1. My first query relates to the description of the feasibility study type: P2 L30: The authors state ‘This will be a pragmatic, randomised controlled...’

Pragmatic trials are designed to test interventions in the full spectrum of everyday clinical settings to maximise applicability and generalisability (e.g. Treweek & Zwarenstein, 2009). Thus, pragmatic in this context suggests the recruitment of any new adult client receiving hearing aids. Yet, the feasibility study (and ensuing trial) limits inclusion based on pure tone audiometric criteria (mild and moderate, not severe or profound). I therefore suggest the authors remove the word pragmatic. The authors should also provide justification for the inclusion criteria limit on PTA within the methods (I expect this is related to the anticipated benefit of the intervention for those two subgroups, based on prior literature?)

We agree with this comment and have accordingly removed the word pragmatic from the text on P2L14 of the revised manuscript. Additionally we have added the following text on P9L2-3:

...since experience 43 suggests this group of patients may struggle to effectively participate in the intervention setting.

2. My second point relates to qualitative analyses: P19 L37: ‘Interviews will be audio recorderd...’ It is not clear who will be responsible for conducting the interviews and analysing/interpreting the data.

We have rewritten this sentence which now reads:

Interviews will be audio-recorded, transcribed verbatim and analysed (with NVivo-11) by the research team led by CJ, using the Framework approach⁶² which is particularly useful for analysing qualitative data in a pragmatic yet systematic way, where theoretical development is not needed.

On P19L17-19.

3. P20, L16: As recruitment is well-underway, are the authors able to specify with more certainty what will be included on their HEA cost-form? i.e. what HE data they are collecting?

We already state on P20L2-3 that “Resource use will be summarised by resource use type (e.g. GP appointments, outpatient appointments)”

We have added to this section on P20L6-14 in the revised manuscript to add some detail and clarity.

“It is anticipated that additional resources utilised in the ACE intervention arm will largely be NHS staff time and travel/time for patients and SOs, patients and SOs use of primary and secondary NHS care, any private treatments attended, whether related to their hearing or for any other reason, changes to medication and employment or recreational activities. A draft resource use questionnaire based on these anticipated additional resources was designed for this study. The questionnaire will be developed further and tested during the feasibility trial to ensure that all relevant and necessary data can be collected to establish a reliable and valid tool with which to capture resource use for a future full economic evaluation”

4. P21, L25: What is the role of the PAP? Explain.

We have provided a short explanation of the role as follows:

The PAP is a group of patients, service users, carers and lay members of the public whose role is to support and advise the Study Management Group on all aspects of the study’s progression and management.

On P21L14-16 of the revised manuscript.

5. P22, L9: ‘Amendments’ – this section is not really needed and could be deleted (if the authors really do wish to keep it, it should appear after ethics approval is stated).

We feel that the amendment section is needed as it reflects the current ISCRTN entry. We have however moved the section as suggested. It now appears on P22L21-P23L6 of the revised manuscript.

6. Final comment – the authors should specify a clear timeline, duration and end-point for the feasibility study, and provide full details of the criteria that will be used to judge whether or not the future RCT is feasible.

We have added a timeline on as suggested, adding the following text:

This study commenced on the 1st February 2017 and is of 4 months duration. The study design is a randomised controlled.....

on P8L1-2 of the revised manuscript.

In addition we have added text providing details of the criteria that will be used to judge whether or not the future RCT is feasible as follows:

The outcomes of this study will inform such a RCT.

The feasibility study will be deemed successful if:

1. 70% of recruitment targets attained for all research components.
2. Study consent/retention rates and proposed sample sizes, indicate delivery of the full RCT is plausible within a 5 year study period.
3. 90% of ACE groups of 5-7 consented participants formed within the intervention window with participants attending 3 of 5 sessions.
4. Economic, acceptability, outcome measure, and fidelity evaluation data successfully collected.
5. Participants, significant-others and audiologists evaluate acceptability of the ACE and RCT positively.

(Measures with over 10% missing data maybe modified/replaced prior to the main trial.)

The criteria for success will result in one of following outcomes:

1. stop: full-scale RCT not be feasible in NHS setting
2. continue: feasible with modifications
3. continue: feasible with no modifications, close monitoring
4. continue feasible as is

on P24L2-17 of the revised manuscript

VERSION 2 – REVIEW

REVIEWER	Dr Helen Henshaw NIHR Nottingham Biomedical Research Centre, University of Nottingham, Otology & Hearing Group, Division of Clinical Neuroscience, School of Medicine.
REVIEW RETURNED	10-Apr-2018
GENERAL COMMENTS	I thank the authors for addressing prior comments clearly and succinctly.